# PDLIM2 Suppression Inhibit Proliferation and Metastasis in Kidney Cancer

**DOI:** 10.3390/cancers13122991

**Published:** 2021-06-15

**Authors:** Hyeong-Dong Yuk, Kyoung-Hwa Lee, Hye-Sun Lee, Seung-Hwan Jeong, Yongseok Kho, Chang-Wook Jeong, Hyeon-Hoe Kim, Ja-Hyeon Ku, Cheol Kwak

**Affiliations:** 1Department of Urology, Seoul National University Hospital, 101 Daehak-ro, Jongno-gu, Seoul 03080, Korea; armenia8@snu.ac.kr (H.-D.Y.); solareclipss@hanmail.net (H.-S.L.); 11shjeong@gmail.com (S.-H.J.); zebra1357@naver.com (Y.K.); drboss@gmail.com (C.-W.J.); hhkim@snu.ac.kr (H.-H.K.); kuuro70@snu.ac.kr (J.-H.K.); 2Department of Urology, Seoul National University College of Medicine, Seoul 03080, Korea; 3Songdo Bio-Engineering, Incheon Jaeneung University, Incheon, 111-15 Songdo-gyoyuk-ro, Yeonsu-gu, Incheon 21987, Korea; Lee12042@snu.ac.kr

**Keywords:** kidney cancer, LIM domain proteins, metastasis, knockout mouse, PDLIM2

## Abstract

**Simple Summary:**

Kidney cancer is a common malignant tumor in both men and women and accounts for approximately 5% of all cancer incidences. Advances in imaging technology and the increasing use of health care facilities have led to the early detection of kidney cancer cases. However, many individuals are still diagnosed with metastatic kidney cancer. If the cancer is accompanied by metastases at the time of diagnosis, the 5-year survival rate is 12%. Despite the beneficial effects of anticancer drug treatment on the survival of patients with metastatic kidney cancer, survival may be less than a year. PDLIM2 plays an essential role in cancer formation and inhibition. To verify oncogenic function of the PDLIM2, we conducted several experiments and animal experiments. Our findings indicating that PDLIM2 may be a new therapeutic target for metastatic kidney cancer.

**Abstract:**

We evaluated the expression of PDLIM2 in human kidney cancer cell lines from primary or metastatic origins and found that PDLIM2 expression was highly elevated in metastatic kidney cancers. We evaluated the effect of PDLIM2 inhibition by RNA interference method. PDLIM2 knockdown showed the decreased proliferation and metastatic character in human metastatic kidney cancer cells. By repeated round of orthotopic injection of RenCa mouse kidney cancer cell line, we obtained metastatic prone mouse kidney cancer cell lines. PDLIM2 expression was highly expressed in these metastatic prone cells comparing parental cells. In addition, we evaluated the in vivo efficacy of PDLIM2 knockout on the tumor formation and metastasis of kidney cancer cells using a PDLIM2 knockout mice. The experimental metastasis model with tail vein injection and orthotopic metastasis model injected into kidney all showed reduced lung metastasis cancer formation in PDLIM2 knockout mice comparing control Balb/c mice. Overall, our findings indicate that PDLIM2 is required for cancer formation and metastasis in metastatic kidney cancer, indicating that PDLIM2 may be a new therapeutic target for metastatic kidney cancer.

## 1. Introduction

Kidney cancer is a common malignant tumor in both men and women and accounts for approximately 5% of all cancer incidences [1]. More than 140,000 people die each year from kidney cancer worldwide, and kidney cancer is the 13th most common cause of cancer death [2]. The number of patients newly diagnosed with kidney cancer has been increasing for decades [3]. Advances in imaging technology and the increasing use of health care facilities have led to the early detection of kidney cancer cases [4]; however, many individuals are still diagnosed with metastatic kidney cancer. The 5-year survival rate of people with kidney cancer is approximately 75%; however, the survival rate depends on a number of factors, including the histologic subtype, tumor size, and stage of cancer upon first diagnosis. Kidney cancer in about two-thirds of patients “diagnosed at early stages is localized to the kidney and these people have a 5-year survival rate of 93%. If kidney cancer has already spread to surrounding tissues or local lymph nodes at the time of diagnosis, then the 5-year survival rate of the patient is 70%. If the cancer is accompanied by distant metastases at the time of diagnosis, the 5-year survival rate is 12% [5]. Despite the beneficial effects of vascular endothelial growth factor inhibitors on the survival of patients with metastatic kidney cancer, survival may be less than a year [5]. Therefore, identification of new molecular targets is necessary for the effective treatment of patients with metastatic kidney cancer.

PDLIM2 plays an essential role in cell differentiation and cytoskeleton formation, and is a member of the actinin-associated LIM family associated with tumorigenesis [6,7]. PDLIM2, located on chromosome 8p21, is a nuclear protein containing both PDZ and LIM domains, with various cellular functions, such as cell migration regulation, cell polarization, and epithelial–mesenchymal transition (EMT). [8] It also regulates the activity of transcription factors, such as NF-κB and STAT, and is associated with the development or inhibition of several malignancies [6,7,9]. The role of PDLIM2 in tumor formation and inhibition is controversial. In many cancers, PDLIM2 levels are epigenetically suppressed by promoter hypermethylation, which blocks transcription. In addition, PDLIM2 upregulation has been linked to a pattern of tumorigenesis suppression [10,11,12]; however, in malignant tumors, the chromosomal region containing the PDLIM2 gene is often destroyed, and high expression of PDLIM2 in cancer cell lines derived from metastatic cancer has been associated with the formation of malignant tumors [13,14]. Therefore, PDLIM2 can play contradictory roles in various tumors and may be a potential therapeutic target. In this study, we evaluated the expression pattern of PDLIM2 in human renal cancer cells and metastatic kidney cancer and the effect of PDLIM2 inhibition on tumor growth and invasiveness.

## 2. Results

### 2.1. Expression of PDLIM2 Gene in Human Kidney Cancer Cell

To investigate the expression of the PDLIM2 gene in renal cancer cells, Western blot analysis, and RT-qPCR analysis were performed on kidney cancer cell cells in the early stage, locally advanced stage, and metastatic stage. PDLIM2 was overexpressed in various stages of human kidney cancer cells. PDLIM2 protein expression was higher in the locally advanced state and the metastatic stage kidney cancer cells than in the early stage (Figure 1A). Additionally, mRNA levels tended to increase toward the locally advanced state, the metastatic stage kidney cancer cells, rather than the early stage (Figure 1B). The analysis of the results of the mRNA sequencing data of TCGA-RCC [15] also showed that the expression of PDLIM2 increased as the stage of RCC was increased, similar to the results of our study (Figure 1C).

### 2.2. Role of PDLIM2 Gene in Human Metastatic Kidney Cancer Cell Proliferation

PDLIM2 was overexpressed in various kidney cancer cell lines of metastatic origin, such as Caki1 and ACHN, compared to those of primary origin (Figure 2A). Based on these results, we hypothesized that PDLIM2 can act as a metastasis-related oncogenic protein in kidney cancer. To confirm whether PDLIM2 acts as an oncogene in human kidney cancer cells, we generated PDLIM2 knockdown stable human metastatic kidney cancer cell lines (shRNA-expressing Caki-1 cell lines; Sh_PDLIM2_01 and Sh_PDLIM2_02) using short hairpin RNA (shRNA) in Caki-1 cells (Figure 2B) and ACHN (Appendix A). We observed that the cell proliferation capabilities of Sh_PDLIM2_01 and Sh_PDLIM2_02 cells in terms of growth rate for up to 72 h were significantly reduced compared to that of the control shRNA (Sh_con) group, the number of cells in which doubled per day (Figure 2C). In addition, clonogenicity, which reflects the self-renewal ability, of Sh_PDLIM2_01 and Sh_PDLIM2_02 cells was significantly reduced compared to that of the control group (Figure 2D).

### 2.3. Inhibition of PDLIM2 Decreases the Migration Ability of Metastatic Kidney Cancer Cells

In the control shRNA group, most cells migrated across the scratch on the culture; however, in the PDLIM2 knockdown groups, Sh_PDLIM2_01 and Sh_PDLIM2_02, many cells did not migrate across the scratch (Figure 3A). In the transwell infiltration assay, the number of cells stained with crystal violet after chemoattraction in the PDLIM2 knockdown groups Sh_PDLIM2_01 and Sh_PDLIM2_02 decreased compared to that in the control shRNA group (Figure 3B). Assessment of the protein and mRNA levels for EMT markers indicated an increase in N-cadherin (CDH2) in the PDLIM2 knockdown group and the concomitant transcriptional and protein upregulation of E-cadherin (CDH1) (Figure 3C,D). Moreover, these results indicate that the specific downregulation of the PDLIM2 gene resulted in phenotypic changes associated with EMT, which impaired the invasion capacity of metastatic kidney cancer cells.

### 2.4. PDLIM2 Expression Is Related to Metastasis Promotion

To investigate the relationship between the increased aggressiveness of cancer cells and PDLIM levels, we performed repeated orthotopic injection of metastatic kidney cancer cells. Primary tumors were formed by direct orthotopic injection of human kidney cancer cells into kidneys. We used RenCa cells expressing GFP/luciferase (pFUGW-luc) for the monitoring of tumor growth. Afterwards, metastatic kidney cancer cells that metastasized to the lungs were collected, cell lines were purified, and metastatic cells were repeatedly and orthotopically injected into kidneys to establish RenCa-M1 to RenCa-M3 cell lines (Figure 4A). We first observed PDLIM2 protein expression in kidney tumors injected with original (M0) and metastatic (M1,M2) RenCa cells. Immunohistochemistry analysis showed that PDLIM2 relatively increased in RenCa-M1 and M2 cells compared to original RenCa cells (RenCa-M0). (Figure 4B). The protein levels of PDLIM in RenCa-M0 and metastatic RenCa cells (RenCa-M1, RenCa-M2 cells) were also detected higher in Western blotting experiment (Figure 4C). The mRNA level of PDLIM2 was significantly elevated in RenCa-M1 to RenCa-M3 cells compared to that in RenCa-M0 cells, and the mRNA levels indicated the tendency of M0 cells to rapidly change into M2 cells (Figure 4D). Similarly, in immunofluorescence staining, PDLIM expression was significantly elevated in RenCa-M1 to RenCa-M3 cells compared to that in RenCa-M0 (Figure 4E). Next, we observed that the cell proliferation capabilities of RenCa-M1 and RenCa-M2 cells for up to 72 h. In terms of growth rate, RenCa-M1 showed slight reduction compared to that of RenCa-M0 cells, but this reduction was recovered in RenCa-M2 and M3 cells (Figure 5A). When we performed wound healing experiment using RenCa M0 to M3 cells, metastatic RenCa cells migrate more across the scratch on the plate. The migration assay indicated the tendency of M0 cells to change into M3 cells (Figure 5B). In the transwell infiltration assay, the number of metastatic RenCa cells stained with crystal violet after chemoattraction increased gradually compared to that of RenCa-M0 cells (Figure 5C). Finally, we compared metastatic characters of M0 to M3 by injecting them in mouse kidney and compared metastatic signals in lung after same time period. Total luciferase signal from the extracted mouse lung after 14 days of kidney injection showed gradual increase in M1 and M2 (Figure 5D). These results indicate that PDLIM2 expression increases in cells with high metastatic tendency and that PDLIM2 is related to metastasis promotion.

### 2.5. Suppression of PDLIM2 Significantly Inhibits Metastatic Tumors in the Tail Vein-Injected Murine Kidney Cancer Model

To verify the relationship the oncogenic function of the PDLIM2 gene in vivo, we established experimental metastasis model via injecting mouse kidney cells expressing GFP/luciferase into the tail vein of control or PDLIM2-knockout mice. The occurrence of lung metastasis after a series of tail vein injections of tumor cells was compared using bioluminescent images. Lung metastasis in PDLIM2-knockout mice was observed to be significantly reduced compared to that in control mice. After the injection, bioluminescence signals from PDLIM2-knockout mice was not increased, whereas signals from control mouse (Balb/c) continued to grow every week (Figure 6A). The bioluminescence signals from the extracted lungs of PDLIM2-knockout mice were found to be significantly reduced compared to that from Balb/c control mice on the day of sacrifice (Figure 6B). Upon hematoxylin-eosin staining of extracted lungs, we found that PDLIM2-knockout mice had a smaller size and fewer cancer sites than the control mice (Figure 6C). Our results indicate that PDLIM2 inhibition can effectively inhibit lung tumor formation in experimental metastasis model of kidney cancer cells xenograft model.

### 2.6. Suppression of PDLIM2 Significantly Inhibits Metastatic Tumors in the Orthotopically Injected Murine Kidney Cancer Model

To verify the relationship between the oncogenic function of the PDLIM2 gene and metastasis in vivo, we established kidney orthotopic tumor model using GFP/lucirease expressing RenCa in control (Balb/c) and PDLIM2-knockout mice. The luciferase signals from each mice were monitored every week (Figure 7A, right graph). Three weeks after orthotopic injection, nephrectomy of primary tumors was performed, and the luciferase signal from the whole mice was calculated through bioluminescence imaging (Figure 7A, left images). Signals from PDLIM2 knockout mice were significantly reduced compared to those from control mice (Figure 7A). After nephrectomy, metastatic cancer growth was monitored through bioluminescence imaging every week (Figure 7B, right graph), and the images from the day of sacrifice were shown (Figure 7B, left images). On the day of sacrifice, the degree of metastasis among the groups was compared by classifying the metastasis sites according to organs, and the metastasis signals in PDLIM2-knockout mice were either relatively small or relatively limited and even absent compared to those in the control mice (Figure 7C). After formaldehyde fixation, liver, spleen, and lung tissues from control and PDLIM2 knockout mice were stained with hematoxylin and eosin (H&E), and we found that the size and the number of the cancers in PDLIM2-knockout mice was smaller than that in the control mice (Figure 7D). Our results indicate that PDLIM2 inhibition can effectively inhibit tumor growth and metastasis in a orthotopic kidney cancer formation model.

### 2.7. Association of Suppression of PDLIM2 and Cancer Formation with Aging

To verify whether the epigenetic genetic repression of PDLIM2 affects tumorigenesis, we observed spontaneous tumorigenicity following aging in PDLIM2-knockout mice and control wild type (Balb/c) mice. Tumor formation was observed in 15-month-old mice, and most of the tumors occurred in the lungs and colon. Unfortunately, we cannot observe kidney cancer formation in this time period. We found PDLIM2 knockout mice had significantly reduced lung tumor formation than wild-type mice. However, PDLIM2 knockout mice had a similar tumor formation rates comparing wild-type mice when observed in the colon (Figure 8A,B). This result further indicates the tissue dependency of tumor formation affected with PDLIM2 expression and moreover, importance of PDLIM2 in metastatic lung cancer formation.

## 3. Discussion

This is the first study to examine the role of PDLIM2 in the tumor growth and metastasis of kidney cancer in a xenograft model. We observed that PDLIM2 is expressed at high levels in metastatic human kidney cancer cell lines comparing non-metastatic kidney cancer cell lines. In particular, the specific inhibition of PDLIM2 was found to result in the reduced proliferation and migration ability of metastatic kidney cancer cells. As metastasis adaptation characteristics were acquired in murine kidney cancer cell line, PDLIM2 expression also was increased. In addition, we provide in vivo evidence that the specific inhibition of PDLIM2 significantly reduces tumor growth and metastasis in a human kidney cancer xenograft model; this is the key finding of our study. Thus, we suggest that PDLIM2 may act as an oncogenic protein in metastatic kidney cancer development and that the selective inhibition of PDLIM2 may be a new therapeutic target for metastatic kidney cancer.

In many previous studies, PDLIM2 has been reported to be a tumor suppressor [11,16,17,18]. PDLIM2 is known to play a role in inhibiting the activation of the tumorigenic factors NF-κB and STAT3 [16]. PDLIM2 acts as a tumor suppressor by inhibiting cancer-related genes and increasing the expression of genes involved in antigen presentation and T-cell activation [16]. Sun et al. reported that PDLIM2 is epigenetically suppressed in lung cancer and is associated with treatment resistance and poor prognosis [17]. In mice, the inhibition of PDLIM2 resulted in increased lung cancer incidence and was reported to cause resistance against anticancer drugs and immunotherapeutic drugs, such as PD-1 inhibitor [17]. In breast cancer, PDLIM2 was associated with adhesion signaling and β-catenin activity; the inhibition of PDLIM2 plays a role in inhibiting tumor growth [18].

However, in our study, higher PDLIM2 expression was observed in the process of in vivo metastasis adaptation of RenCa kidney cancer cells, and we found that PDLIM2 is highly related to tumor growth and metastasis in mouse knockout model, The epigenetic genetic repression of PDLIM2 negatively affects growth and expression of kidney cancer and spontaneous tumorigenesis following aging. These results suggested that it has a role as an oncogenic protein.

In a cancer genome atlas analysis, PDLIM2 was reported to be an unfavorable prognostic factor in kidney cancer. In pathological tissues, high PDLIM2 expression was reported to be linked to poor prognosis [19]. The principle underlying the role of PDLIM2 in tumor growth and metastasis is not yet clear. In previous studies, we reported that PDLIM2 is associated with the mitogen-activated protein kinase (MAPK)/extracellular signal-regulated protein kinase signaling pathway (ERK) and that an increase in PDLIM2 is associated with the activation of this pathway [10]. In addition, PDLIM2 was suggested to interact with the MAPK/ERK signaling pathway to regulate MET and various genes, such as cell cycle proteins [20]. MAPK and mitogen-activated kinases (MKKs) play an essential role in several biological pathways that regulate cell differentiation, proliferation, and survival [20]. Activated in response to extracellular stimuli, MKK phosphorylates ERK and MAPK [20]. The MAPK/ERK signaling pathway plays an essential role in cell proliferation and differentiation and in tumor formation and metastasis [20]. The MAPK/ERK signaling pathway can show both oncogenic and tumor-suppressive effects depending on the tissue-specific tumor microenvironment. In addition, mutations affecting the MAPK/ERK pathway depend on the type of cancer. These differences in tissue and tissue-specific tumor microenvironment may have influenced the role of MAPK/ERK signaling pathway and PDLIM2 gene. The epigenetic genetic repression of PDLIM2 have influenced the MAPK/ERK pathway depending on the tissue-specific tumor microenvironment.

Bowe et al. also reported that PDLIM2 is highly expressed in invasive cancer cells [21]. In a previous study, we found PDLIM2 to be expressed at higher levels in castration-resistant prostate cancer (CRPC)-like cells in prostate cancer [10]. Moreover, we confirmed that the specific inhibition of PDLIM2 resulted in significantly reduced tumor growth in human CRPC xenograft models. The inhibition of PDLIM2 resulted in a reduced number and weakened viability, proliferation, and clonal growth of prostate cancer cells; furthermore, PDLIM2 may play a carcinogenic role in human CRPC-like cells [10]. The expression level of PDLIM2 in androgen-sensitive prostate cancer cells and CRPC-like cells was significantly different, and PDLIM2 was expressed at a high level in CRPC-like cells [10]. This suggests that PDLIM2 may play several roles in regulating tumor growth and progression in various organs and circumstances. Bowe et al. reported that PDLIM2 is highly expressed in invasive cancer cells [19]. Additionally, the inhibition of PDLIM2 has been reported to reduce the expression of several oncogenes and increase tumor suppressor gene expression [19].

The incidence, metastasis, and angiogenesis of several malignant tumors, including kidney cancer, have been reported to be associated with the activation of the MAPK signaling pathway [22]. Angiogenesis is essential for kidney cancer. Vascular endothelial growth factor inhibitors are commonly used against metastatic kidney cancer targets, such as angiogenesis-related targets. MAPK/ERK activation is associated with angiogenic growth factor signaling in tumor cells, suggesting that PDLIM2 may act as an oncogenic protein that promotes tumor formation and metastasis by activating MAPK/ERK [10]. Furthermore, Huang et al. reported that the inhibition of the MAPK and ERK pathways in kidney cancer markedly inhibits the growth of anchorage-independent RCC cells, reduces the extension of tumor cells, and reduces tumor angiogenesis. Therefore, the inhibition of the MAPK signaling pathway leads to the destruction of tumor vasculature [22].

In addition, PDLIM2 may contribute to the ability of tumor cells to migrate [10,19,23]. In the present study, a decreased migration ability of metastatic kidney cancer cells was observed in the PDLIM2 knockdown group, in which PDLIM2 was selectively inhibited. The PDLIM2 gene encodes a member of the ALP subfamily of proteins in the PDZ-LIM domain [23]. The encoded protein inhibits anchorage-dependent growth and promotes cell migration and adhesion by interacting with the actin cytoskeleton [23]. Bowe et al. reported that the inhibition of PDLIM2 inhibited the reversal of the EMT phenotype, loss of orientation, and cytoskeleton polarization [19].

The preclinical data of this study showed that the regulation of PDLIM2 expression, which regulates the MAPK/ERK signaling pathway, is involved in tumor growth and proliferation in metastatic renal cancer. In addition to the tumor, it was found that the regulation of PDLIM2 expression of the individual was also involved in the growth and proliferation of the tumor. Furthermore, the epigenetic genetic repression of PDLIM2 influenced spontaneous development of tumors. PDLIM2 has an essential regulatory role that may alter the response of cancer cells, and current evidence shows that PDLIM2 is a viable target for cancer therapy. However, further studies are needed to determine the applicability of PDLIM2-targeted treatment in actual clinical practice.

Our study has some limitations. First, we observed an association between PDLIM2 and carcinogenesis. There is only partial in vitro evidence for the oncogenic signaling pathway associated with PDLIM2. Second, to utilize our results in clinical settings, it is necessary to confirm whether PDLIM2 is highly expressed in the kidneys of patients with kidney cancer, whether there is a difference between primary kidney cancer and metastatic kidney cancer in terms of PDLIM2 expression, and whether PDLIM2 is correlated with the prognosis of kidney cancer.

## 4. Materials and Methods

### 4.1. Materials

Dulbecco’s modified Eagle’s medium (DMEM), trypsin, antibiotics, TRIzol, and Lipofectamine 2000 were purchased from Invitrogen (Carlsbad, CA, USA). Fetal bovine serum and culture media were obtained from HyClone Laboratories Inc. (South Logan, UT, USA). Anti-PDLIM2 antibodies were purchased from Novus (Littleton, CO, USA) and Abcam (Cambridge, UK). Antibodies against ECAD and NCAD were obtained from Cell Signaling Technology (Danvers, MA, USA). Anti-β-actin antibody (Cat. No. A2066) and other chemicals were purchased from Sigma-Aldrich (St. Louis, MO, USA).

### 4.2. Cell Lines, Plasmids, Virus Production, and Infection

HK-2, ACHN, and 7860 cell lines were purchased from American Type Culture Collection (Manassas, VA, USA). Caki-1, RenCa, and 293T cell lines were purchased from Korean Cell Line Bank (Seoul, Korea). HK-2 cells were cultured in Keratinocyte SFM medium (GIBCO, Carlsbad, CA, USA). Caki-1, ACHN, 786O, and RenCa cells were cultured in MEM (WELGENE, Gyeongsan-si, Korea), and 293T cells for lentiviral packaging were cultured in DMEM (WELGENE, Gyeongsan-si, Korea) supplemented with 10% fetal bovine serum at 37 °C under 5% CO_2_. For gene silencing, control and PDLIM2 shRNA-expressing lentivirus infected and stable cell lines were established as previously described [15]. The oligonucleotide sequences for the PDLIM2 shRNA are listed in Appendix A. The FUGW-luc vector was obtained from the Molecular Imaging and Neurovascular Research Laboratory, Dongguk University Ilsan Hospital, Goyang, Republic of Korea; FUGW-luciferase-expressing cells were produced as follows. The FUGW-luc vector was cut using the XhoI enzyme and transfected into the RenCa cell line (RenCa-GFP). The cells incorporated with FUGW-luc were sorted with the GFP channel in BD FACSAria II (BD Biosciences, Franklin Lakes, NJ, USA).

### 4.3. Colony Formation Assay and Cell Viability Assay

For the colony formation assay, 1000 cells/well were plated in 6-well plates. The cells were cultured for 14 days and then stained with 0.1% crystal violet. The cell colonies were photographed, and the number of colonies comprising more than 50 cells was counted using a SZX7 stereo microscope (Olympus, Tokyo, Japan). For the cell viability assay, cells (2000 to 3000 cells/well) were dispensed in 100-μL culture medium in a 96-well plate and incubated for certain periods. Then, a 10 μL solution of the EZ-Cytox cell viability kit (Daeil-Lab, Seoul, Korea) was mixed with the culture medium in each well of the plate. Samples were incubated for 1 h at 37 °C, and the absorbance of each sample at 450 nm was measured using a microplate reader (PerkinElmer, Waltham, MA, USA).

### 4.4. RNA Isolation and Real-Time Quantitative Polymerase Chain Reaction (RT-qPCR)

Total cellular RNA was extracted using TRIzol reagent (Ambion, Austin, TX, USA) according to the manufacturer’s instructions. For each reverse-transcription reaction, 1 μg total RNA was used for cDNA synthesis using the MultiScribe Reverse Transcription Kit from Life Technologies (Carlsbad, CA, USA). RT-qPCR was performed using the EvaGreen qPCR Master Mix Kit from Applied Biological Materials Inc. (Richmond, BC, Canada) and the StepOne™ Real-Time PCR System (Applied Biosystems, Foster City, CA, USA). The quantity of 18S ribosomal RNA was measured as an internal control. The sequences of the primers used for reverse transcription and qPCR are listed in Appendix A.

### 4.5. Western Blotting

The cells (5 × 10^6^) were lysed in 1 mL RIPA buffer (150 mM NaCl, 50 mM Tris-HCl [pH 7.2], 0.5% NP-40, 1% Triton X-100, and 1% sodium deoxycholate) containing a protease/phosphatase inhibitor cocktail (Sigma-Aldrich, St. Louis, MO, USA). The cell lysates were separated on sodium dodecyl sulfate-polyacrylamide gels and transferred onto an Immobilon-P PVDF Membrane (Millipore, Darmstadt, Germany). The membranes were blocked with 5% skim milk in 0.1% Tween-20 for 1 h, followed by overnight incubation at 4 °C with primary antibodies. The membranes were incubated with a horseradish peroxidase-conjugated secondary antibody (1:5000) for 1 h and developed using the ECL-Plus Kit (Thermo Scientific, Rockford, IL, USA).

### 4.6. Wound Healing and Cell Invasion Assays

The wound healing assay was performed on 100% confluent cells plated in 6-well culture plates. The cells were scratched using a pipette tip and washed twice to remove debris; fresh medium was then added. The cells were incubated in a humidified atmosphere containing 5% CO_2_ at 37 °C and observed using a SZX7 stereo microscope at certain time points. After the indicated time period, the scratched areas were measured using the ImageJ program (ver. 1.43u; www.rsb.info.nih.gov/ij, accessed on 10 December 2020). For the invasion assay, cells (5 × 10^4^/well) were plated in the upper chambers of transwell culture plates without serum using Matrigel-coated polycarbonate membranes (Corning, Big Flats, NY, USA). Basal medium containing 10% fetal bovine serum was added to the lower chambers as a chemoattractant for cell migration. After 48 h, the cells that did not migrate (non-migrating cells) were removed from the upper chambers, whereas the cells that migrated through the chambers were fixed using 10% ethanol (Sigma-Aldrich, St. Louis, MO, USA). Afterwards, the cells were stained with 0.01% crystal violet solution (Sigma-Aldrich), and the cells that migrated were randomly counted across five different microscopic fields at 20× magnification.

### 4.7. Animal Studies

All animal experiments were performed in accordance with the Seoul National University Hospital institutional guidelines under IACUC protocol No.19-0058-S1A0. PDLIM2 knockout mice were kindly provided by Dr. Takashi Tanaka (RIKEN Laboratory, Kanagawa, Japan). To establish a blood-borne metastatic model, 1 × 10^6^ RenCa-GFP cells in 100 μLPBS were injected into the tail vein of six-week-old BALB/c or PDLIM2 knockout mice (*n* = 8 for each group). Bioluminescent signals were measured every week for 4 weeks; the lungs with tumors were excised, and bioluminescent signals were measured. The lungs with tumors were photographed and fixed with paraformaldehyde (PFA). The tumor specimens were stained with H&E. To establish a metastatic model with orthotopic tumors, 1 × 10^5^ RenCa-GFP cells expressing a luciferase expression vector were injected into the kidneys of six-week-old male BALB/c mice or PDLIM2 knockout mice (*n* = 5 for each group). For the injection step, cells were suspended in 100 µL of 50% Matrigel (BD Biosciences) in complete medium. Bioluminescent signals were measured every week for 3 weeks until the nephrectomy of tumor-bearing kidney was performed. After nephrectomy, metastatic signals were measured for another 3 weeks. The mice were euthanized, and organs were extracted, measured for bioluminescent signals, fixed in 4% PFA at 4 °C, and embedded in paraffin. Then, the tumor specimens were stained with H&E.

### 4.8. In Vivo Bioluminescence Imaging

The mice from each group were intraperitoneally injected with 150 mg/kg VivoGlo^TM^ D-luciferin (Promega, Madison, WI, USA) and imaged after 15 min. After anesthetizing the mice using 1–3% isoflurane, the photons emitted from the tumor were detected using Xenogen IVIS Imaging System 200 (Alameda, CA, USA). Depending on the experimental conditions, the settings for the target max count (3000 or 30,000 photon counts) and exposure time (10 s. or auto-exposure with a maximum exposure time of 1 min) were applied for imaging. Regions of interest (ROI) of the same size and shape were used for all acquired images to measure the total flux (photons per sec) in the ROI. Living Image (Version 2.20, Xenogen, Alameda, CA, USA) was used to quantify signals emitted from the ROI.

### 4.9. Generation of the Metastasis-Prone Adapted RenCa Cells

Metastasis-prone adapted RenCa cells were established as follows. RenCa-GFP cells were injected into 6-week-old male BALB/c mouse kidneys. Primary tumors were monitored weekly using bioluminescence signals, and 3 weeks after injection, the tumor-bearing kidneys from each mouse were removed; metastatic signals were monitored for another 3 weeks. On the day of euthanasia, the tumor masses in the lungs were surgically dissected under sterile conditions and incubated with trypsin for 2 h at 37 °C. The dissociated cells were selected in vitro by adding zeocin under normal cell culture conditions for 1 week. These metastatic cells were considered as metastatic round 1 (RenCa-M1), and the expanded tumor cells were re-implanted into the mouse kidney. This in vivo metastatic cycle was performed two more times until highly metastatic RenCa cells were established (M2,M3).

### 4.10. Immunohistochemistry

Kidney tumors arising from orthotopically injected original (M0) and metastatic RenCa cells (M1,M2) were fixed using 4% formaldehyde solution overnight. After dehydration, the tissues were embedded in paraffin. Serial sections were sliced, mounted on silanized glass slides, and stained with H&E. The tumor slides were immune-stained with PDLIM2 antibody.

### 4.11. Statistical Analyses

All data were analyzed using Microsoft Excel 2010 software, unless otherwise stated. Normally distributed continuous variables were analyzed using Student’s *t*-test. All the statistical tests were two-tailed. *p* < 0.05 was considered statistically significant.

## 5. Conclusions

PDLIM2 is highly expressed in highly metastatic kidney cancer. The inhibition of PDLIM2 decreases the invasion capacity and proliferation of kidney cancer cells and the production of crones. Here, PDLIM2 inhibition resulted in the inhibition of tumor growth in an in vivo metastatic kidney cancer xenograft model, indicating that PDLIM2 may be a new therapeutic target for metastatic kidney cancer.

## Figures and Tables

**Figure 1 cancers-13-02991-f001:**
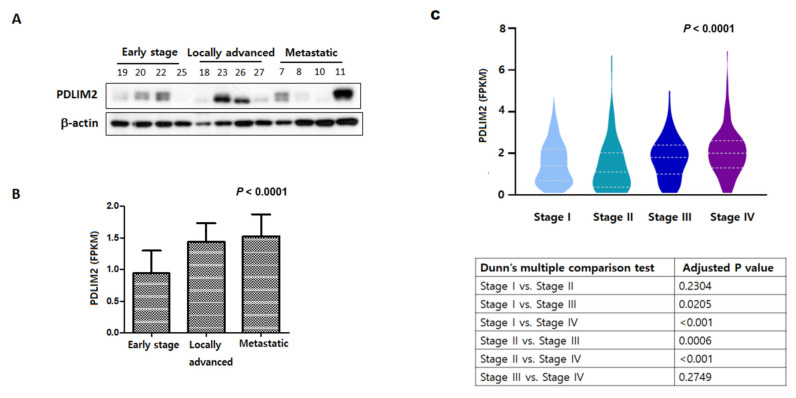
Expression of PDLIM2 gene in human kidney cancer cell. (**A**) PDLIM2 protein expression in various renal cancer cells. Whole cell lysates were analyzed using the indicated antibodies. (**B**) The mRNA levels of various renal cancer cells. Total RNAs from each cell line were measured using RT-qPCR. (**C**) The results of the mRNA sequencing data of TCGA-RCC and analysis.

**Figure 2 cancers-13-02991-f002:**
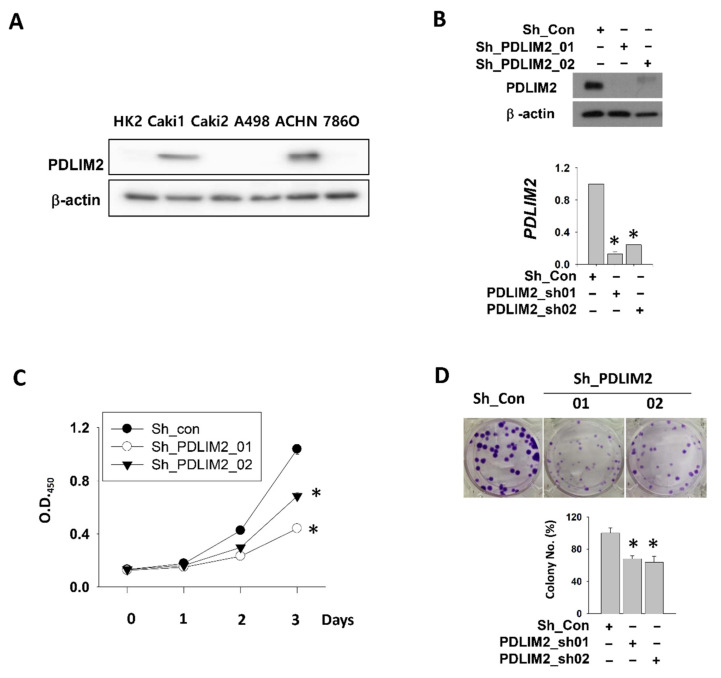
PDLIM2 inhibition attenuates metastatic kidney cancer cell proliferation. (**A**) PDLIM2 protein expression in various kidney cancer cell lines. Whole cell lysates were analyzed using the indicated antibodies. (**B**) The efficiency of PDLIM2 knockdown was measured by comparing the protein and mRNA levels of PDLIM2 in an shRNA-expressing Caki-1 cell line. Whole cell lysates were analyzed using the indicated antibodies (upper figure). Total RNAs from each cell line were measured using RT-qPCR (lower graph). (**C**) Time-dependent viability changes of control and PDLIM2 shRNA-expressing Caki-1 cells were assessed using EZ-Cytox solution. Bars represent the mean ± standard deviation (SD) of three independent experiments, and * denotes *p* < 0.05 (Student’s *t*-test) versus the control shRNA (Sh_con) group. (**D**) Crystal violet staining images of colonies formed by a similar number of shRNA-expressing Caki-1 cells. The average number of colonies counted is shown in the lower graph. Bars represent the mean ± SD of three independent experiments, and * denotes *p* < 0.05 (Student’s *t*-test) versus the control shRNA (Sh_con) group.

**Figure 3 cancers-13-02991-f003:**
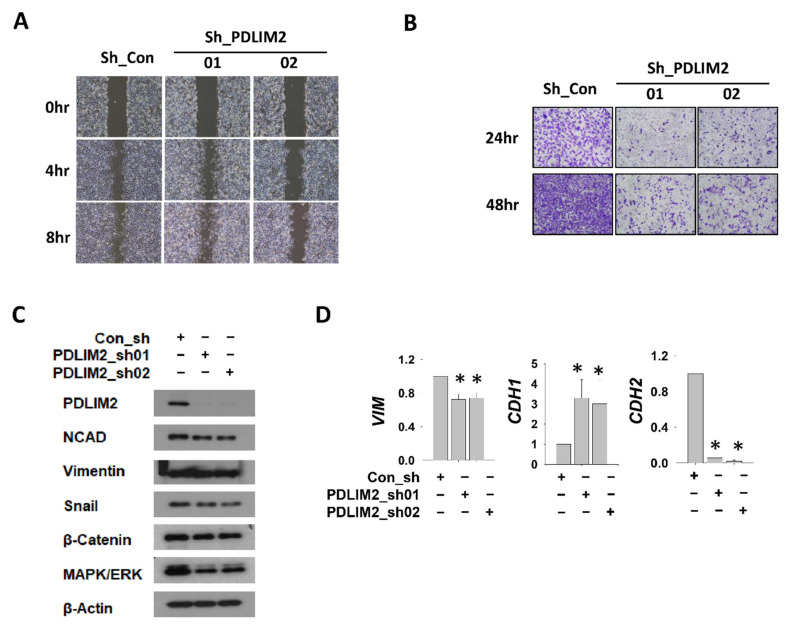
PDLIM2 knockdown reduced cell mobility and EMT-related gene expression in metastatic kidney cancer cell lines. (**A**) Cell migration after scratch-wounding control and PDLIM2 shRNA-expressing Caki-1 cells was observed after certain time points. (**B**) Transwell infiltration assay of stable cells with same cell numbers. At an indicated time point after plating, the cells that had migrated to the underside of the filters were fixed and stained with crystal violet, and photographs were captured. (**C**) The protein levels of the indicated EMT markers in the whole cell extracts of the control and PDLIM2 knockdown cells were measured. (**D**) The mRNA levels of indicated EMT marker genes in control Caki-1 cells and PDLIM2 knockdown Caki-1 cells were measured. Bars represent the mean ± standard deviation (SD) from three independent experiments, and * denotes *p* < 0.05 (Student’s *t*-test) versus the sh-con group.

**Figure 4 cancers-13-02991-f004:**
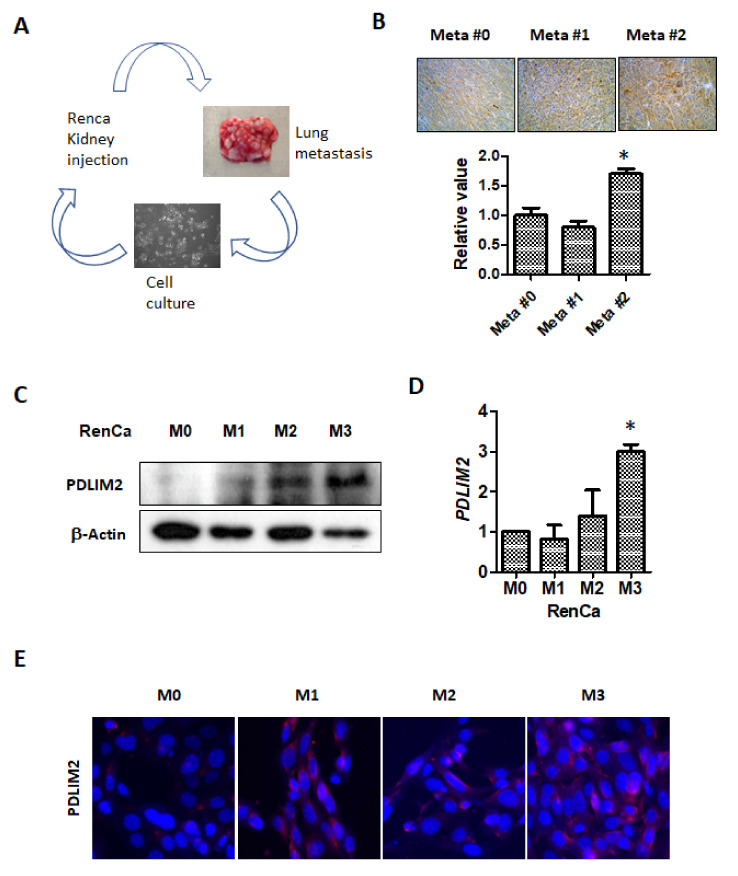
PDLIM2 levels were elevated in metastatic murine kidney cells. (**A**) Schematic of metastatic RenCa-M1 cells transforming into RenCa-M3 cells. (**B**) Immunohistochemistry images of PDLIM2 expression in the indicated orthotopic tumors of metastatic RenCa cells. Three different areas from each image were quantitated using the Image J program and plotted. Bars represent the mean ± standard deviation (SD) of three independent experiments, and * denotes *p* < 0.05 (Student’s *t*-test) versus the M0 group. (**C**) The protein levels of PDLIM2 in control and metastatic RenCa cells were measured using whole cell extracts. (**D**) The mRNA levels of PDLIM2 in the indicated cells were measured using RT-qPCR. Bars represent the mean ± SD of three independent experiments, and * denotes *p* < 0.05 (Student’s *t*-test) versus the sh-con group. (**E**) Immunofluorescence staining of PDLIM2 in original (M0) and metastatic RenCa (M1–M3) cells.

**Figure 5 cancers-13-02991-f005:**
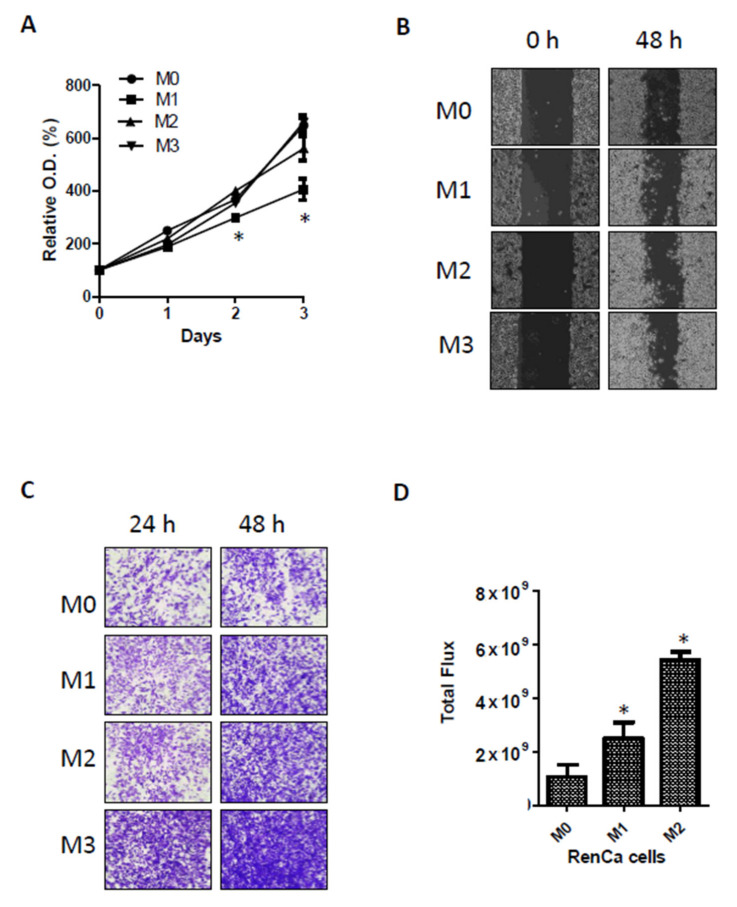
Characteristic of metastatic prone RenCa cells. (**A**) Cell growth rate of original RenCa (M0) and metastatic RenCa (M1–M3). Time dependent viability changes of M0–M3 RenCa cells were assessed using EZ-Cytox solution. Bars represent the means ± SDs of three independent experiments, and * denotes *p* < 0.05 (Student *t*-test) versus the M0 group. (**B**) Scratch-wounding cell migration of the M0–M3 RenCa cells were observed after indicated time. (**C**) The trans-well infiltration assay of the same number of indicated RenCa cells. At indicated time after plating, cells that had migrated to the underside of the filters were fixed, stained with crystal violet and photographs were taken. (**D**) Bioluminescent flux plot from the live mice injected with M0-M2 after 14 days of injection. Error bars represent means ± SEM of each group (*n* = 3). * *p* < 0.05 (Student’s *t*-test) versus the M0 group.

**Figure 6 cancers-13-02991-f006:**
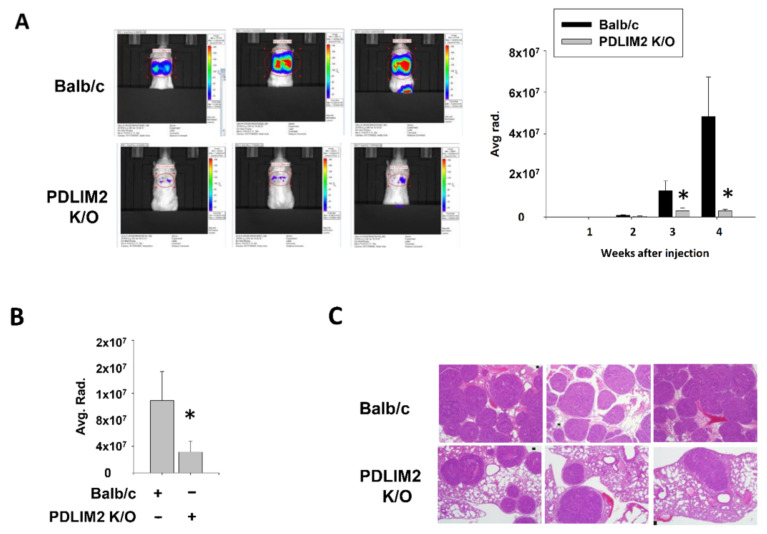
PDLIM2 knockout mice showed decreased metastatic tumors in tail vein injected model of RenCa murine kidney cancer cells. (**A**) Bioluminescent images from the indicated mice at the day of sacrifice. Time dependent bioluminescent flux plot quantifying tumors are graphed left. Error bars represent means ± SEM of each group (*n* = 10). * *p* < 0.05 (Student’s *t*-test) versus the control Balb/c group. (**B**) Bioluminescent flux plot from the extracted lung from each group. Error bars represent means ± SEM of each group (*n* = 10). * *p* < 0.05 (Student’s *t*-test) versus the control Balb/c group. (**C**) Representative hematoxylin-eosin staining images of the lung metastasis tumors.

**Figure 7 cancers-13-02991-f007:**
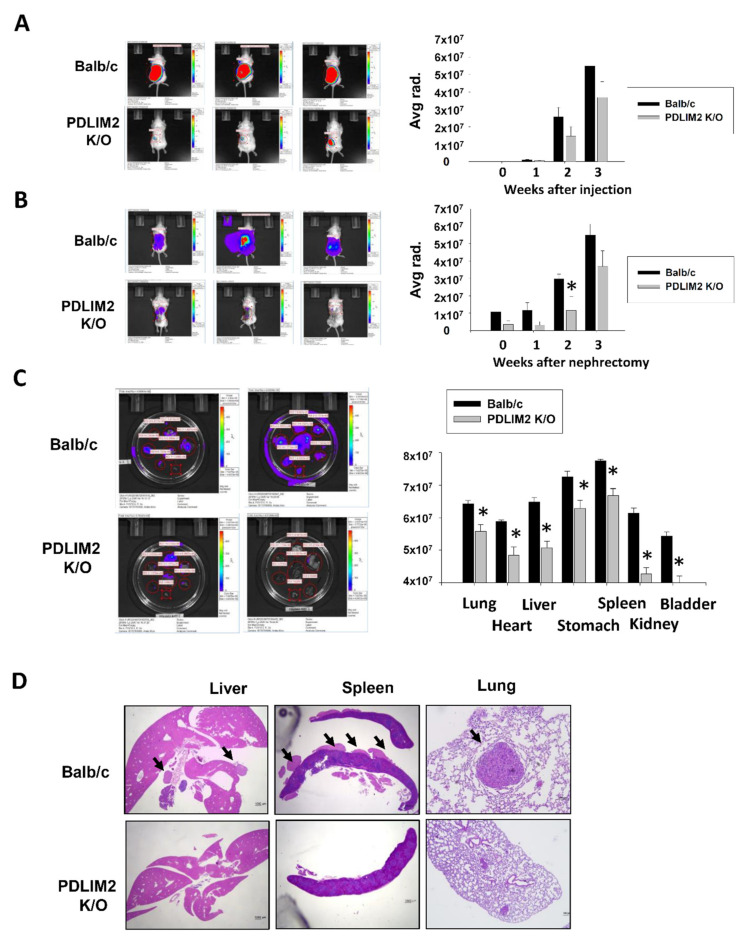
PDLIM2 knockout mice showed metastatic tumors in kidney orthotopic injected model of RenCa murine kidney cancer cells. (**A**) Bioluminescent images from the indicated mice at the day of nephrectomy. Time dependent bioluminescent flux plot quantifying tumors are graphed left. Error bars represent means ± SEM of each group (*n* = 10). * *p* < 0.05 (Student’s *t*-test) versus the control Balb/c group. (**B**) Bioluminescent images from the indicated mice at the day of sacrifice. Time dependent bioluminescent flux plot quantifying tumors are graphed left. Error bars represent means ± SEM of each group (*n* = 10). * *p* < 0.05 (Student’s *t*-test) versus the control Balb/c group. (**C**) Bioluminescent images from the indicated mice organ at the day of sacrifice. Average bioluminescent flux plot quantifying each tumors are graphed left. Error bars represent means ± SEM of each group (*n* = 10). * *p* < 0.05 (Student’s *t*-test) versus the control Balb/c group. (**D**) Representative hematoxylin-eosin staining images of the metastasis tumors from indicated mice. Arrow indicates tumor area.

**Figure 8 cancers-13-02991-f008:**
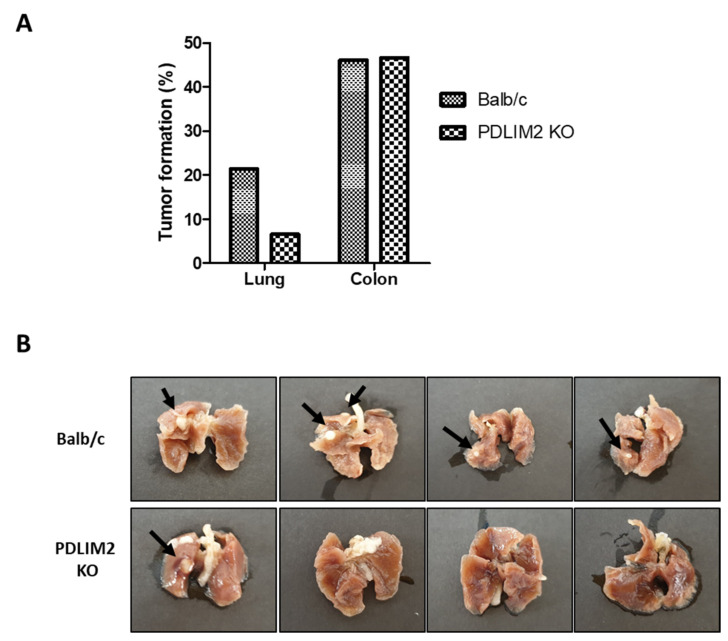
Lung cancer formation upon aging was reduced in PDLIM2 knockout mice. (**A**) Tumor formation rate in the lung and colon of 15-month-old wild-type (Balb/c) and PDLIM2 knockout (KO) mice: Balb/c (*n* = 13), PDLIM2 KO (*n* = 15). (**B**) Images of formaldehyde-fixed lungs extracted from indicated mice. Black arrows indicate lung tumors.

## Data Availability

The data presented in this study are available on request from the corresponding author. The data are not publicly available due to security issues.

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
