# Peer review of "PDLIM2 Suppression Inhibit Proliferation and Metastasis in Kidney Cancer"

_cancers, 2021, doi:10.3390/cancers13122991_

Round 1

Reviewer 1 Report

Most of my comments seem to have been addressed. There are few issues that should probably be addressed. 

  1. The PDLIM2 KO mice experiments haven't been properly rationalized and explained. If the view was to look for tumor cell extrinsic effects of the KO and how it might effect metastasis of injected cells this hasn't been clearly stated. Also all the mechanistic studies and discussion talk about tumor cell intrinsic role of the gene in metastasis. This experiment however suggests a function of the gene in the TME, which is never mentioned or a likely method ever discussed.
  2. PDLIM KO has different effect on tumor development in various organs. Have the author looked to check if their KO is to the same extent in various tissues? Can the difference in spontaneous tumor development be defined by difference in level of the gene's KO in various tissues and consequent differences in activation of MAPK-MEK signaling?

Author Response

Thank you for your letter and the helpful comments regarding our manuscript. We were pleased to note the favorable comments. We have carefully reviewed their comments and made the necessary changes to the original manuscript, which are indicated by blue font in the revised manuscript.

Reviwer1

The PDLIM2 KO mice experiments haven't been properly rationalized and explained. If the view was to look for tumor cell extrinsic effects of the KO and how it might effect metastasis of injected cells this hasn't been clearly stated. Also all the mechanistic studies and discussion talk about tumor cell intrinsic role of the gene in metastasis. This experiment however suggests a function of the gene in the TME, which is never mentioned or a likely method ever discussed.

▶ Thanks for this comment. We revised the sentences in discussion in line in line 338, page 14 and line 354, page 14

PDLIM2 may play tumor suppressor or oncogenic roles in regulating tumor growth and progression in various organs and circumstances. PDLIM2 is associated with the mitogen-activated protein kinase (MAPK)/extracellular signal-regulated protein kinase signaling pathway (ERK) and that an increase in PDLIM2 is associated with the activation of this pathway. The MAPK/ERK pathway demonstrates both oncogene and tumor suppressor effects depending on the tissue-specific tumor microenvironment. The epigenetic genetic repression of PDLIM2 was thought to affect the MAPK/ERK pathway depending on the tissue-specific tumor microenvironment. We determine the effect of this epigenetic genetic repression of PDLIM2 on the growth and proliferation of kidney cancer.

PDLIM KO has different effect on tumor development in various organs. Have the author looked to check if their KO is to the same extent in various tissues? Can the difference in spontaneous tumor development be defined by difference in level of the gene's KO in various tissues and consequent differences in activation of MAPK-MEK signaling?

Thanks for this comment.We revised the sentences in results and discussion, in line 303, page 13and line 338, page 14To verify whether the epigenetic genetic repression of PDLIM2 is just a bystander event or actually a driver of tumorigenesis, we examined whether PDLIM2 genetic deletion leads to development of spontaneous tumors,The MAPK/ERK pathway demonstrates both oncogene and tumor suppressor effects depending on the tissue-specific tumor microenvironment. Also,mutations affecting MAPK/ERK pathways differ depending on the type of cancer. This difference in tissue and carcinoma may have influenced the difference in spontaneous growth in knock-out mice.

Reviewer 2 Report

The study show several data suggesting the role of PDLIM2 into the pathogenesis of meatastatic behaviour in renal cancer, showing convincing in vitro and in vivo studies.

The week part of the study is still the limited muber of patients'specimens used, that as generally happens show a quite large variability. 

Thus, I would further stress in the discussion section that preclinical data suggest the possible effect of targeting this protein to treat metastatic kidney cancer expressing PDLIM2. However, additional studies are required to show whether this is a real possibility in the clinical setting. 

Author Response

Thank you for your letter and the helpful comments regarding our manuscript. We were pleased to note the favorable comments. We have carefully reviewed their comments and made the necessary changes to the original manuscript, which are indicated by blue font in the revised manuscript.

Reviwer2

The study show several data suggesting the role of PDLIM2 into the pathogenesis of meatastatic behaviour in renal cancer, showing convincing in vitro and in vivo studies.

The week part of the study is still the limited muber of patients'specimens used, that as generally happens show a quite large variability.

Thus, I would further stress in the discussion section that preclinical data suggest the possible effect of targeting this protein to treat metastatic kidney cancer expressing PDLIM2. However, additional studies are required to show whether this is a real possibility in the clinical setting.

▶ Thanks for this comment. We added the sentences in discussion, in line 389, page 15

This manuscript is a resubmission of an earlier submission. The following is a list of the peer review reports and author responses from that submission.

Round 1

Reviewer 1 Report

The authors present a data here that suggests that PDLIM2 plays a oncogeneic role in kidney cancers. The present expression profiling, in-vitro assays and in-vivo experiments to put across this point. Though interesting I find that the study has several issues that need to be addressed before this manuscript can be considered for publication:

  1. Fig 1A: The western blot actually shows fait bit of variability in expression of PDLIM2 across stages of cancers. There is also no statistical analysis presented making drawing conclusions from this figure improper and possibly faulty. Also the figure lacks a legend making interpreting both Fig 1A and B difficult.
  2. Fig 2A the cell ines aren't annotated I'm not sure which cell line corresponds to which tumor stage. 
  3. Fig 2C shows growth effects of KO of PDLIM2, the data is presented in a single cell line so making generalized conclusions from here seems like a long shot. Similarly for the other phenotypic assays like colony formation etc. Experiments in a single cell-line can be confounded by cell-line specific effects. 
  4. Fig 3C ECAD blot isn't very clear. 
  5. "The migration assay indicated the tendency of M0 cells to change into M3 cells" - It's unclear how this assays give the authors any information about tumor evolution. 
  6. The experiments in KO animals perplex me. How do these experiments tell us anything about the intrinsic biology of PDLIM2 in the injected tumor cells. At best these experiments tell us about how expression of PDLIM2 in normal tissues effect engraftment and migration. Also the fact that the KO mice never form kidney tumors even at very old age strongly indicates to me that PDLIM2 isn't itself oncogenic but might rather promote growth and migration after tumor development or it's over-expression is a passenger event. 
  7. The authors also extensively discuss MAPK\ERK signaling in the context of PDLIM2 in the discussion but none of the data even attempts to show how this pathway is perturbed in the current experiment models. I detailed examination of this pathway is warranted given its importance. 

Author Response

Thank you for your letter and the helpful comments regarding our manuscript. We were pleased to note the favorable comments. We have carefully reviewed their comments and made the necessary changes to the original manuscript, which are indicated by blue font in the revised manuscript.

1)  Fig 1A: The western blot actually shows fait bit of variability in expression of PDLIM2 across stages of cancers. There is also no statistical analysis presented making drawing conclusions from this figure improper and possibly faulty. Also the figure lacks a legend making interpreting both Fig 1A and B difficult.

▶ Thanks for this comment. We added a statistical analysis, legend, and Figure 1C. Figure 1C is the Results of mRNA sequencing data fr0m TCGA-RCC. We added the sentences in line142, page4

2) Fig 2A the cell ines aren't annotated I'm not sure which cell line corresponds to which tumor stage. 

▶ Thanks for this comment. Annotated which cell line corresponds to which tumor stage. In line 162, Page5

3) Fig 2C shows growth effects of KO of PDLIM2, the data is presented in a single cell line so making generalized conclusions from here seems like a long shot. Similarly for the other phenotypic assays like colony formation etc. Experiments in a single cell-line can be confounded by cell-line specific effects. 

▶ Thanks for this comment. The results of experiments conducted on the ACHN cell line are attached to Supplemetary figure 1.

4) Fig 3C ECAD blot isn't very clear. 

▶ Thanks for this comment. We rerun the western blot and corrected Figure 3C. ECAD protein does not seem to be well expressed in western blot in RCC cell line. Only trends could be identified in qPCR. We added PCR results in ACHN cell line are attached to Supplemetary figure 2. 

5) "The migration assay indicated the tendency of M0 cells to change into M3 cells" - It's unclear how this assays give the authors any information about tumor evolution. 

▶ Thanks for this comment. We added the sentences, in line 223, page 7

6) The experiments in KO animals perplex me. How do these experiments tell us anything about the intrinsic biology of PDLIM2 in the injected tumor cells. At best these experiments tell us about how expression of PDLIM2 in normal tissues effect engraftment and migration. Also the fact that the KO mice never form kidney tumors even at very old age strongly indicates to me that PDLIM2 isn't itself oncogenic but might rather promote growth and migration after tumor development or it's over-expression is a passenger event. 

▶ Thanks for this comment. First of all, through an experiment, we tried to find out the role of PDLIM2 in the tumor itself, and in KO mice, we also wanted to see the effect of PDLIM2 on the microenvironment of the individual during the tumor formation process.

7) The authors also extensively discuss MAPK\ERK signaling in the context of PDLIM2 in the discussion but none of the data even attempts to show how this pathway is perturbed in the current experiment models. I detailed examination of this pathway is warranted given its importance. 

▶ Thanks for this comment. We added MAPK/ERK in Figure 3

Reviewer 2 Report

The manuscript is a novel study showing the potential impact of targeting PDLIM2 signalling to treat metastatic kidney cancer. The hypothesis is intriguing and data are quite convincing. I have only few remarks that may help to address whether these experimental studies may be translated into clinical utility.

1) Figure 1 shows a wide variability of PDLIM2 protein expression among patients. A larger number of individuals to be analyzed should be important to confirm the increased levels in advanced disease stati  or to address the reason of variability among patients.

2) To test whether PDLIM2 suppression reverses EMT and to analyze its dowstream signalling pathway (with the final aim to detect potential therapeutic targets modulating this pathway) , it would be of interest to address  the epithelial marker E-cadherin and the transcriptional regulators of EMT, as Snail and β-catenin and/or its downtream effectors, as shown in other cancer models (Bowe et al, Mol Biol of cell 2013). Addressing the molecular mechanism underlying the hypothesis of PDLIM2-dependent metastatic expansion would be of help. 

Author Response

Thank you for your letter and the helpful comments regarding our manuscript. We were pleased to note the favorable comments. We have carefully reviewed their comments and made the necessary changes to the original manuscript, which are indicated by blue font in the revised manuscript.

1) Figure 1 shows a wide variability of PDLIM2 protein expression among patients. A larger number of individuals to be analyzed should be important to confirm the increased levels in advanced disease stati  or to address the reason of variability among patients.

▶ Thanks for this comment. We added a statistical analysis, legend, and Figure 1C. Figure 1C is the Results of mRNA sequencing data fr0m TCGA-RCC. We added the sentences in line142, page4

2) To test whether PDLIM2 suppression reverses EMT and to analyze its dowstream signalling pathway (with the final aim to detect potential therapeutic targets modulating this pathway) , it would be of interest to address  the epithelial marker E-cadherin and the transcriptional regulators of EMT, as Snail and β-catenin and/or its downtream effectors, as shown in other cancer models (Bowe et al, Mol Biol of cell 2013). Addressing the molecular mechanism underlying the hypothesis of PDLIM2-dependent metastatic expansion would be of help. 

▶ Thanks for this comment. We added Snail and β-catenin in Figure 3
